# Atomistic Study for the Tantalum and Tantalum–Tungsten Alloy Threshold Displacement Energy under Local Strain

**DOI:** 10.3390/ijms24043289

**Published:** 2023-02-07

**Authors:** Mohammad Bany Salman, Minkyu Park, Mosab Jaser Banisalman

**Affiliations:** Virtual Lab Inc., 38 Wangsimni-ro, Seongdong-gu, Seoul 08826, Republic of Korea

**Keywords:** tantalum–tungsten, threshold displacement energy, molecular dynamics calculation, DFT, radiation damage, strain

## Abstract

The threshold displacement energy (TDE) is an important measure of the extent of a material’s radiation damage. In this study, we investigate the influence of hydrostatic strains on the TDE of pure tantalum (Ta) and Ta–tungsten (W) alloy with a W content ranging from 5% to 30% in 5% intervals. Ta–W alloy is commonly used in high-temperature nuclear applications. We found that the TDE decreased under tensile strain and increased under compressive strain. When Ta was alloyed with 20 at% W, the TDE increased by approximately 15 eV compared to pure Ta. The directional-strained TDE (E_d,i_) appears to be more influenced by complex 〈*i j k*〉 directions rather than soft directions, and this effect is more prominent in the alloyed structure than in the pure one. Our results suggest that radiation defect formation is enhanced by tensile strain and suppressed by compressive strain, in addition to the effects of alloying.

## 1. Introduction

During the operation of a nuclear reactor, collisions between incident energetic particles and the atoms in the structural materials can lead to the formation of defects due to atomic displacement. These defects can have significant impacts on the material’s properties, such as increased hardening, a shift in the ductile to brittle transition temperature and decreased thermal conductivity. It is therefore important to accurately estimate the types and amounts of irradiation defects formed in order to understand the negative effects on the structural materials of the nuclear reactor.

Tantalum (Ta) and its alloys are among the materials that are considered for use as constituent materials in nuclear reactors. These materials have received significant attention due to their potential use in high-temperature nuclear applications, such as space shuttle reactors [1]. In the late 1950s and early 1960s, several alloying combinations of Ta were investigated to develop refractory alloys with improved mechanical and thermal properties, in which solid solution (W and Re additions) alloys were the focus of most of this effort. While Ta-based alloys have a higher density than Nb-based alloys, for example, they have inferior low-temperature density-compensated strength. Ta has a higher melting temperature (3290 K), resulting in better strength retention and density-compensated creep strength above 1000 K [1]. Although this alloy exhibited a suitably low ductile to brittle transition temperature (DBTT), it has good workability and good weldability [2]. It has a body-centered cubic (BCC) lattice structure and a density of 16.65 g/cc at ambient temperature. At room temperature, Ta is extremely ductile.

Early examination of the creep substructure in Ta-2.5 wt%W reveals the existence of interaction chances between moving dislocations and dispersion particles [3,4]. The formation and evolution of void and dislocation arrangements in Ta, Ta-5 wt.%W and Ta-10 wt.%W was assessed as a function of low radiation level at a temperature of 345 ± 3 °C, [5], in which W delays the loop evolution and therefore the formation of radiation-induced voids. Furthermore, it was found that the addition of 10 wt.%W allows for the retention of appropriate nonirradiated ductility and weldability [6].

During the nuclear reactor operation, however, materials are subject to various sources of strain, including irradiation-induced swelling, alloying/solute precipitates, void swelling, and solute segregation. Void-metal interactions sometimes become extremely high [7], and deuterium precipitates exhibit strain fields of up to 5% in one direction due to local volume expansion [8]. Moreover, the structural material of the nuclear reactors will be subjected to combined heat and mechanical stresses. These stressors affect the stable defect type, defect density, and defect distribution, hence impacting the lifetime of the materials. In fact, earlier molecular dynamics (MD) studies of collision cascades have demonstrated that stress affects the number of defects created by collision cascades in several refractory materials, including W, Mo, Fe, and V [9,10]. A study of point defect distributions in Ta was carried out by simulating displacement cascades initiated with PKA atom energies ranging from 5 to 20 keV [11]. The plastic deformation mechanisms in shock-wave compression of single-crystal Ta were approached considering different propagation directions and force-filed potential models [12]. Operational conditions such as strain were also among the studied work for the Ta alloys. The stress–strain dynamic response and deformation mechanism of Ta–W alloy have been studied for the potential use in military applications [4].

For evaluating the radiation damage and defect production of materials, the threshold displacement energy (TDE) is a key value to measure. TDE represents the minimum kinetic energy required to displace an atom from its lattice site into a defect position, and is the first step in the formation of point defects such as vacancies and interstitials [13]. It is commonly known that the formation of a stable Frenkel pair (FP), which is a combination of self-interstitial atoms (SIA) and vacancies, requires an energy greater than the TDE. Thus, TDE is used to predict the radiation tolerance of nuclear materials and to identify materials with high resistance to radiation damage [14]. Additionally, TDE can be used to predict the microstructure, mechanical and thermal properties of materials exposed to radiation, which is important for understanding the effects of radiation on materials performance and reactor safety and efficiency. Theoretically, the NRT (Norgett, Robinson, Torrens) method can be used to estimate the number of defects based on the evaluated TDE [15].

The average TDE over a high enough number of displacement directions of several refractory materials, including W, Mo, and Fe, has been previously evaluated using molecular dynamics (MD) simulations under both strained and non-strained conditions [16]. To the best of our knowledge, however, there have been no studies to date that have used MD simulations to estimate the TDE of Ta under either strained or non-strained conditions.

This study aims to evaluate the average TDE of Ta under both non-strained and strained conditions, as well as to assess the potential effects of alloying Ta with W on TDE. To do so, we performed MD simulations on pure Ta and Ta alloys containing 5–30% W at intervals of 5% W content. The simulations were conducted to investigate the effect of strain on the averaged TDE of these materials.

## 2. Results

### 2.1. The Free Strained TDE

#### 2.1.1. The Evaluation for Pure Ta

To validate the current EAM potential model, the results of key point defects characteristics of W in bcc bulk Ta measured with DFT and compared to MD are summarized in Appendix A. The substitutional formation energy of W in Ta bulk is −0.15 eV with DFT. The two substitutional 1st and 2nd-nearest neighbor of W-W atoms (i.e., 1NN, 2NN) were investigated. The 1NN and 2NN binding energy are −0.135 eV and −0.048 eV, respectively. The negative sign means that the W-W substituted pairs show repulsive interaction in the Ta structure. Furthermore, from Figure 1, it is clearly shown that both Ta-SIA, and Ta–W mixed dumbbell pairs have the lowest formation energy at <111> dumbbell. Meanwhile the Ta–W <111> dumbbell has lower formation than that of the <111> of pure Ta-SIA. This tells us that the Ta–W interstitial is more stable than the Ta-SIA. In both MD and DFT cases, the Ta–W <111> dumbbell has the lowest formation energy. It is also important to mention that the <111> dumbbell also had the lowest energy among the other configurations when the structure was W-Ta [17,18].

With MD simulation, the minimum TDE, i.e., E_d,min_, was confirmed to occur over the 〈100〉 direction (26.5 eV), which was also confirmed experimentally to be the lowest TDE value for the same direction with a E_d,min_ = 25 eV [20]. Furthermore, in another temperature dependence study, this direction was found to exhibit a minimum TDE experimentally [21].

However, it can be challenging to obtain TDE results for directions other than 〈100〉 experimentally due to the technical difficulty of setting the dislocation direction accurately along the 〈111〉 or 〈110〉 direction. Even slight angle deviations from these exact directions can impact the TDE results.

To demonstrate the impact of angle deviations from specific collision directions on TDE values, we analyzed the effect on E_d,i_ values for recoils in typical directions, namely 〈100〉, 〈110〉, 〈111〉, and 〈321〉 (Figure 2).

The current MD-EAM potential model shows that the 〈100〉 direction of E_d,i_ exhibits the most stable and slowest deviation changes for angles up to 14° (Figure 2). This model reasonably reproduces experimental results on the resistivity changes induced by electron irradiation as a function of the incident electron energy. Experimentally, the TDE of Ta was found to be around 36 eV in a region of 20° around the 〈111〉 direction (the current MD value is around 38 eV), and it was approximately 53 eV in a region of 18° around the 〈100〉 direction [22] (the current MD value is approximately 48 eV). TDE values were larger than 130 eV for other directions, including 〈110〉 and 〈111〉, with the current MD value showing large variation from 30 eV to 150 eV for complex directions. Experimentally, the average of E_d,i_ over the three directions 〈100〉, 〈110〉, and 〈111〉 has been found to be 32 eV for Ta [20]. However, this value may underestimate the overall E_d,avg_, as other complex directions have been shown to have higher E_d,i_ values.

Additionally, theoretical evaluation of TDE, for example using Jan and Seeger equation, has been shown to inaccurately predict E_d,avg_ in the case of bcc- Fe [23]. Such theoretical evaluation of TDE was conducted for W by Mason [24], and per his calculation, the E_d,avg_ of W = 55.3 eV. In a similar manner, to further validate our study, we have conducted a compative study between our TDE evaluation over 210 directions (using the theoretical model as described by Mason) and our MD results. The finding was similar to the result reported in [25] for W. The theoretical equation assumes that (I) E_d,i_ changes smoothly around the 〈100〉, 〈110〉, and 〈111〉 directions. However, as seen in Appendix A, the pattern around a direction such as <321> deviates from this assumption, leading to significant underestimations of E_d,i_ values for many directions in the theoretical estimates, and thus an underestimation of E_d,avg_ for Ta.

These findings suggest that it is challenging for experiments to accurately determine TDE values for directions other than <100> due to the sensitivity of E_d,i_ to small deviations in the recoil direction. The response of TDE to angle deviation has previously been reported for other body-centered cubic materials, such as tungsten [26] and molybdenum [27]. Analysis of the angle deviation factor between our MD results and experimental data supports the validity of the current potential model for the directional evaluation of TDE. Overall, to properly estimate the TDE, E_d,i_ must be averaged over 210 directions and four simulation timings of each direction, i.e., the evaluated E_d,avg_ is equal to 61 eV for pure Ta.

#### 2.1.2. The Evaluation of TDE for Ta–W Alloy (Effect of W Contents)

As compared to the pure Ta, the alloying of Ta with W seems to prompt changes in the physical and structural characteristics of the Ta materials [28], hence values such as TDE evaluation would be affected. As per our selection for the W ratios range from (5%~30%) of the TDE evaluated with 210 ICD (Figure 3). The error bars represent the average standard error of the mean (SEM) of the TDE across six different configurations of the W ratio, calculated as SEM $=\frac{\sigma }{\sqrt{n}}$, where σ is the standard deviation and n is the number of independent simulations (210 directions and 4 different timings for each direction). The SEM error bar is typically around ±3~4 eV for the TDE.

We found that 20% of W with Ta alloying results in the largest shift of the TDE increment for the Ta–W alloy (TDE = 75.9 eV). Ta alloying with W appears to enhance the TDE of defect formation, possibly because higher W content increases structural strength. Ta alloying with 10% tungsten (Ta-10 W) improves high-temperature strength, since the melting temperature rises as the W ratio increases. It was previously reported that when the W alloying ratio increased, the melting point temperature increased [28].

It is worthwhile to comprehend the basis for the variance in TDE evaluations caused by W alloying. Ta, the periodic table’s next precursor to W, should have a similar band structure to W, but with a lower Fermi level resulting from the absence of one electron [29]. Furthermore, W and Ta have relatively close cohesive energies, with W having the highest value compared to any transition metals in the periodic table [30,31]. Its inter- and trans-granular fracture characteristics may be influenced by this fact. The interatomic bonding strength properties of W or Ta are related to the electronic aspects of these bcc structures, and the fact that W has an approximately “half-filled” 5 d band with the Fermi energy located at the pseudo-minimum of the electronic DOS, as compared to Ta [32]. The existence of the pseudo-minimum allows bonding and antibonding states classification formed by 5-d orbitals (Appendix A). The Fermi energy shifts toward the minimum of DOS as the concentration of W increases, and bonding in the alloys becomes more covalent and less metallic. Although the trend of 5% W is relatively like that of pure Ta, the 5% alloyed structure could exhibit more metallic properties than the pure Ta structure, which may explain why this ratio has the lowest TDE. The metallic bond shall be weaker as compared to the covalent bond that could explain lower the TDE resulting from the 5% ratio.

It is worth noting that the miller indices’ complex type of displacement directions influences the TDE evaluation. In Figure 4, the directions about 〈100〉 (the least complicated) seem to be less affected by the alloying factors as well, implying that this route resulted in reasonably close TDE for both pure Ta and Ta–W alloys. By contrast, most complex directions, such as the direction around 〈321〉 seems to be significantly affected by W alloying. In Figure 4b, the directions around 〈321〉 and 〈432〉 (highly complex) had the largest TDE and the hottest region. The alloyed structure shows the highest change response for these directions compared to the pure Ta, where the TDE difference in Figure 4c (TDE_Ta–W_–TDE_Ta–W_) clearly confirmed these hot directions around 〈321〉 and 〈432〉. In general, the PKA has a higher chance of colliding with a greater number of surroundings atoms, once traveling in a more complex direction such as 〈321〉 (see Appendix A). When the PKA moves in a complex direction, the energy will be dissipated into a variety of collisions, reducing the likelihood of a head-on collision that would transfer the most energy and cause a defect to occur. (See Appendix A).

### 2.2. Strained TDE

$E_{d,avg}^{strained}$ is the average of $E_{d,i}^{strained}$ over a sufficiently large set of displacement directions. In terms of strain effects, the Ta and Ta–W alloys respond similarly to the strain applications: $E_{d,avg}^{strained}$ decreases with strain when moving from compression to tension (Figure 5). Similar findings were reported in previous studies for pure refractory materials such as bcc-W, Mo, V [33]. It is worth noting that the alloyed structure of Ta–W appears to be stable under this strain level, with the trend close to that of pure Ta but with a higher TDE value for the alloyed structure.

The higher the local tensile strain, the lower the FP formation energy, and thus the lower the TDE of defect creation. The FP formation energy (most stable SIA defect and vacancy formation energy) clearly demonstrated this correlation for Ta. (See Appendix A). Where the trend would not change with the strain’s type, when comparing a hydrostatic strain to other strain such as a uniaxial strain, the hydrostatic strain has similar trends, but its TDE changes more than that of the uniaxial strain (Appendix A); in other words, the uniaxial strain has a lower influence on the $E_{d,avg}^{strained}$ compared with hydrostatic strain for same amplitude.

### 2.3. Effects of Strain on the Directional Dependent TDE

To determine the strain effects on TDE in correlation with a specific displacement direction, six different directions, shown in Figure 6, were simulated for hydrostatic strain in Ta. Sets of directions differing in their [*ijk*] indices are listed in (d1)–(d6). Each data point was obtained by averaging the results of 20 different simulations’ timings to reduce the statistical error and to properly understand the displacement direction effect on $E_{d,i}^{strained}$. The calculated results for each direction under the applied deformations are expressed with the TDE change rate (E_CR_ %) =$\;\frac{E_{d,i}^{strained}-E_{d,i}^{0}}{E_{d,i}^{0}}\times 100$, obtained with reference to the free strained E_d,i_ ($E_{d,i}^{0}$). In Figure 6, least E_CP_ is observed for the 〈111〉 direction, and it was less responsive when the structure is compressed. In general, the compression strain leads for positive E_CP_ and negative E_CP_ when it is tensile strained. The E_CP_ clearly decreases with deformation when moving from compression to tension, although its amplitude is dependent on the displacement direction.

To further analyze the direction dependence of E_CR_, the results over 210 directions are shown in Figure 7. The ECRs are calculated for hydrostatic deformations in Ta, and for Ta–W, the results for the 1.4 % compression strain in (Figure 7a,c) and for the 1.6% tensile strain (Figure 7b,d) are shown. Some anisotropy is observed. In Figure 7a,c, the displacement directions close to the 〈111〉 miller indices are relatively weakly affected by the strain, which could be a result of the focused collision sequence. In the focused collision sequence, atoms sequentially collide in one direction, a pattern of which would not be significantly affected by a change in the number density of atoms. In Figure 7b,d, even with tension, the TDE decreases around the 〈432〉 direction, as indicated by the grey color for Ta, where this effect is strongly shown for Ta–W alloy in black color. However, as a general trend, from Figure 6 and Figure 7, E_CR_ % become negative for a tensile strain while becoming positive for a compression strain in most directions.

## 3. Methods

### 3.1. TDE Calculation for Pure Ta

The non-strained TDE is commonly defined as an average value of the threshold displacement energy in a specific direction, *i,* (*E_d_*_,_*_i_*_,_) over a proper number of displacement directions (Equation (1)). The average value is called *E_d,avg_*. Similarly, we define the strained TDE as an average value of $E_{d,i}^{strained}$, which is the threshold displacement energy for the strained structure in the specific direction *i*. Thus, $E_{d,avg}^{strained}$ is the average value determined in a system under strain conditions.
(1)$$E_{d,avg}=\frac{1}{N_{\mathrm{ICD}}}\sum_{i}^{N_{\mathrm{ICD}}}E_{d,i\;}=\frac{1}{N_{\mathrm{ICD}}}\sum_{i}^{N_{\mathrm{ICD}}}\left\{\frac{1}{N_{\mathrm{timing}}}\sum_{i}^{N_{\mathrm{timing}}}{\;E}_{d,i,j}\;\right\}{\;}_{.}$$
where E_d,i,j_ is the threshold energy determined for the i-th direction recoil at the j-th timing simulation, and E_d,i_ is the averaged threshold energy determined for the i-th direction recoil.

To accurately estimate the average displacement energy (*E_d,avg_*), it is generally necessary to consider many displacement directions. In this study, we obtained a set of nearly uniformly distributed displacement directions by randomly generating points on a unit sphere and converting their coordinates into displacement directions that satisfy the conditions x > y > z and x ≥ 0, y ≥ 0, z ≥ 0. This method produces a list of irreducible crystal directions (ICDs), which correspond to the directions obtained from the origin to the final positions of the points. For example, in our calculations, we used 10,000 uniformly distributed directions to obtain approximately 210 ICDs, which is approximately 1/48 of the total number of original directions. This approach for creating a set of nearly uniform recoil directions has been previously described in an earlier work. [34]

All MD simulations are performed using the large-scale atomic/molecular massively parallel simulator (LAMMPS) code [35] integrated in the materials-square platform [36]. In order to appropriately estimate the TDE, the present study adopts the tested and validated calculation settings from our previous work for example the system size, time step, and PKA energy increments [25]. The simulation cell of 8 × 8 × 12 BCC supercell containing 1536 atoms was created using the structuring tools integrated in materials-square GUI [36], and the initial conditions for the non-strained structure were set to an equilibrium temperature of 30 K and pressure of 0 Pa.

The initial temperature of the recoil simulation is generally set to 30 K, but in certain circumstances it is set to 0 K. The former is referred to as the 30 K recoil simulation, whereas the latter is referred to as the 0 K (to obtain data free of the influence of atom thermal vibration). The goal of the 30 K recoil simulation is to carry out simulations at a temperature close to that of the early experiments [22].

The thermal vibration effect, which is the variation in TDE estimates due to atomic vibrations at a temperature of 30 K [25], was addressed in our simulations by performing multiple runs for each displacement direction. Specifically, we used four simulations to evaluate *E_d,avg_* and $E_{d,avg}^{strained}$ at each direction. We set the interval between recoil events to 50 fs and started the simulation at different times: 0, 50, 100, and 150 fs. The average of the results from these simulations help to minimize the influence of the thermal vibration and time correlations effect in the TDE results.

The interatomic interactions are addressed using the embedded atom method (EAM) potentials based on F-S formalism that were initially characterized by Chen et al. [17]. The potential parameters of formation energy and migration energy of for Ta–Ta were determined by fitting to a set of experimental and first-principles data and the W–Ta cross parameters were fitted to the set of first principles data.

Every recoil event was given an adaptive time step with a maximum displacement (x_max_) of 0.01 Å per step, and a maximum time step (t_max_) of 0.002 ps. After further examination, we can conclude that these values are appropriate for E_d,i_ evaluation when compared to different x_max_ and t_max_ settings. The MD simulation was performed for around 5 ps, and we use the Voronoi analysis implemented in the VORONOI package [37] in the LAMMPS code to determine whether a defect has formed. If there are two or more atoms in the lattice site, a self-interstitial atom (SIA) is created; otherwise, if it is empty, a vacancy is formed. If the lattice site has only one atom, the structure is intact, and no defects are created.

Each recoil event was begun by applying recoil energy to an atom positioned near the lattice’s center; this atom was known as the primary knock-on atom (PKA). Once introduced, the recoil energy was expressed in the velocity components of the PKA. The recoil energy was initially set to 10 eV, then increased by 1 eV until a defect was detected. By using this method, we will calculate the minimum TDE for free-strained and strained structures.

### 3.2. TDE Calculation for Ta–W Alloy

To create a Ta–W alloy structure, the set number of randomly chosen Ta atoms in the supercell were replaced by W atoms to create a system with certain W content. Thus, several Ta–W systems with W content ranging from 5% up to 30% with an interval of 5% were built. Different random distributions were generated by changing the randomly chosen Ta atoms. In the process of simulation, six different random distributions were generated for each W content system to enlarge the statistical size when the structure is non-strained. In conclusion, when different configurations of a same W concentration were examined, the resulting average value can be regarded as being based on quasi-random structures. The PKA was chosen as a Ta atom that was placed at the center of the simulation box. So, for each ratio, the TDE evaluated by obtaining (six different random W ratios) × 210 (ICD), could guarantee the reliability of the results statistically [38]. We also considered the vibration effects, in which for each displacement direction, four simulations were performed, i.e., the TDE of each system with a specific W content ratio was obtained by averaging the results of 5040 simulations. Each simulation consisted of 6 × 210 displacement directions and four starting times for the recoil events. The evaluation of TDE for alloyed structures will be conducted using the method outlined in Equation (2).
(2)$$E_{d,avg,@alloy}=\frac{1}{N_{s}}\sum_{1}^{N_{s}}E_{d,avg},$$

Ns represents the number of unique, randomly generated structures for each W content in the Ta–W alloy.

### 3.3. Strain Application

In practice, high strains such as −1.4% and 1.6% would develop locally around substantial lattice defects such as voids and grain boundaries [7], as well as when impurities such as hydrogen [8] and helium accumulated during reactor operation.

In this study, the hydrostatic strain was applied to all surfaces of the simulation cell. The simulated cell was deformed in the [1 0 0], [0 1 0], and [0 0 1] directions at a deformation rate of 0.002 ps^−1^ up to the targeted strain under the NVT ensemble until the strain value was achieved. This resulted in a total of six distinct simulation cells that were analyzed.

### 3.4. DFT Calculation

DFT calculations using the Quantum ESPRESSO code [31,39] that is integrated in materials square platform [36] are performed to relax atomic structures and investigate electronic properties. An energy cutoff is obtained using projector augmented wave (PAW) pseudopotentials [40] and the Perdew–Burke–Ernzerhof (PBE) generalized gradient approximation (GGA) [41] exchange-correlation potentials. A (4 × 4 × 4) supercell of body-centered cubic (BCC) Ta including 128 atoms is adopted in our simulations. Electron states are expanded in plane waves with an energy cut Ecut(wfc = 60), Ecut(rho = 600), and the Monkhorst–Pack scheme k-points (3 × 3 × 3) are adopted. The present studies start by determining the ground state bulk lattice parameters of Ta in the BCC structure (3.308 Å) which is in good match with the experimental value 3.306 Å [42]. For all cases (bulk of pure and alloys) atoms and volume are allowed to relax (full geometry optimization) until the total force on each atom is smaller than 0.01 eV/Å.

## 4. Conclusions

We have studied the threshold displacement energy (TDE) of pure and alloyed Ta for both non-strained and strained conditions. We wanted to reveal the strain effect on one of the design parameters values for Ta sustainability once utilized in irradiated environments. Applying the molecular dynamic simulation tools, we observed that when the strains ranged from 1.4% compression to 1.6% tension, the TDE decreased from compression to tension on both the pure Ta and Ta–W alloys. Additionally, the effect was more prominent when the structure is alloyed, and 20% of W alloying led to a 15 eV increment for the reported TDE. For the displacement directional dependence of the strain effect, the directional strained TDE (*E_d_*_,_*_i_*) seems to be more influenced by the complex direction rather than the soft direction, and this effect was more prominent in alloyed structure than the pure one.

## Figures and Tables

**Figure 1 ijms-24-03289-f001:**
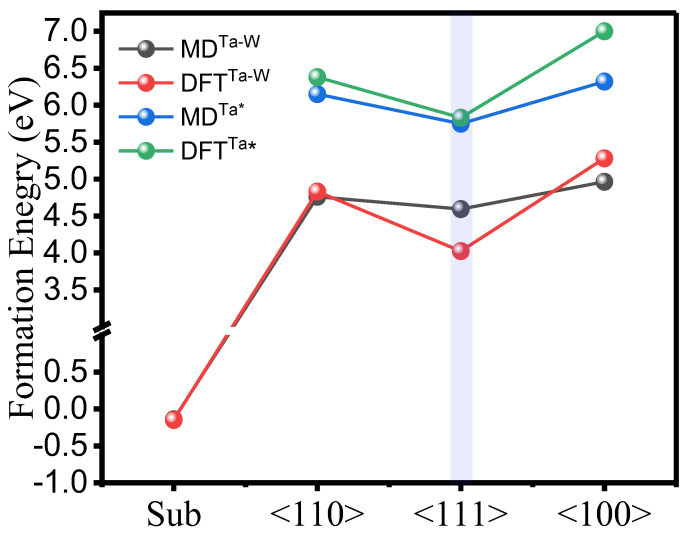
A comparative study of the W substitutional (sub) and the Ta–W mixed dumbbell interstitial (〈110〉, 〈111〉, and 〈100〉) formation energies using the current EAM (MD) study potential model with the density functional theory (DFT) calculations at free strain. The DFT results of pure Ta are obtained from ref. [19], while the results of MD are obtained from [17] in (*).

**Figure 2 ijms-24-03289-f002:**
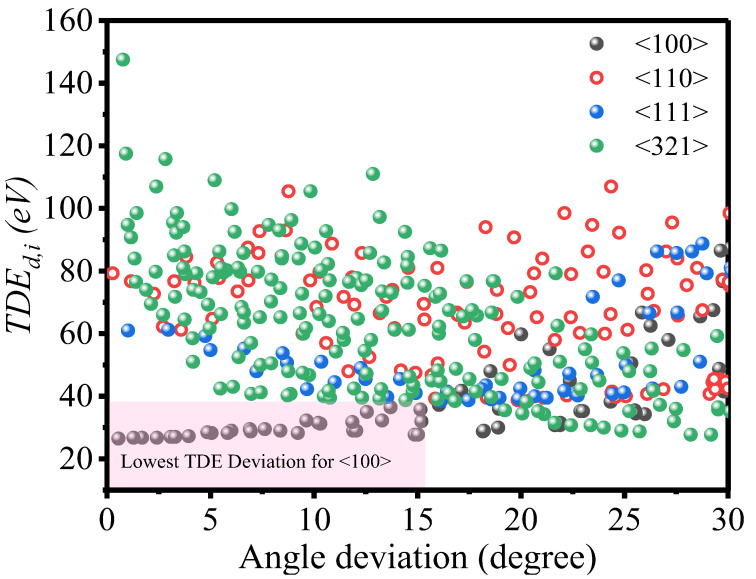
Illustrates how the direction deviation affects the E_d,i_ values of 〈100〉, 〈110〉, 〈111〉 and 〈321〉 recoils. For 〈100〉 recoil, the E_d,i_ values slowly and smoothly change up to around 14° deviations. On the other hand, in other directions, especially for 〈110〉 and 〈321〉 recoils, the E_d,i_ values change strongly even with a slight deviation.

**Figure 3 ijms-24-03289-f003:**
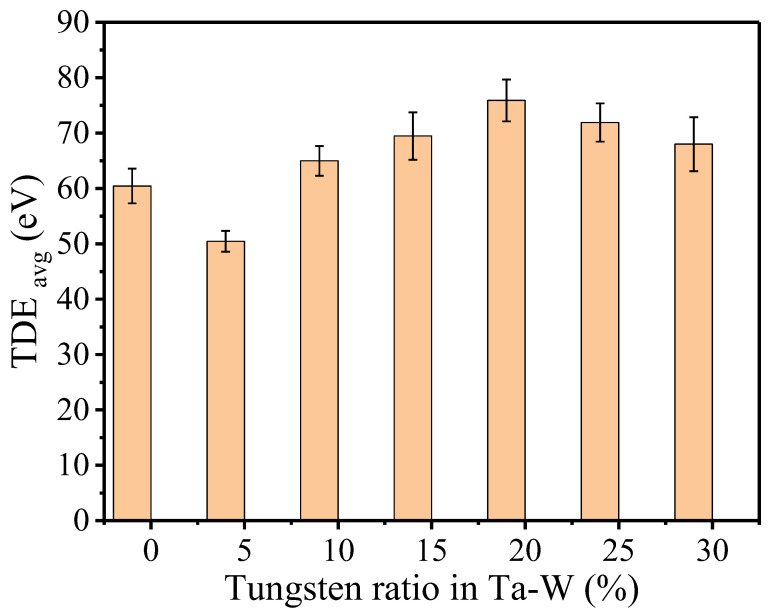
The results of TDE averaged over 210 ICD, for four different vibrational timings and for six different structures at specific W % contest in Ta–W alloy.

**Figure 4 ijms-24-03289-f004:**
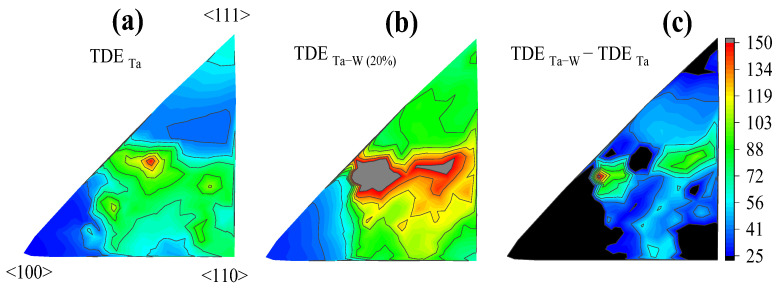
Contour plot of the three-dimensional surface of E_d,i_. The MD results of E_d,i_ over 210 different directions, presented for (**a**) the pure Ta (**b**) Ta–W (20%) alloy (**c**) the energy difference between (**a**), and (**b**). This graph shows that directions around 〈321〉 or 〈423〉 have been strongly affected by W alloying; the highest level of TDE increment stated for with these directions, while the lowest was for <100>.

**Figure 5 ijms-24-03289-f005:**
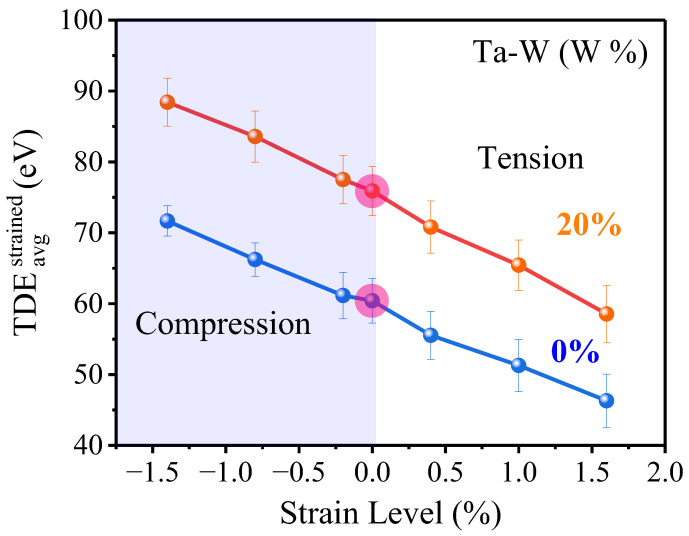
${\mathrm{TDE}}_{i}^{\mathrm{strained}}$, as functions of the strain from compression to tension for pure Ta and Ta–W.

**Figure 6 ijms-24-03289-f006:**
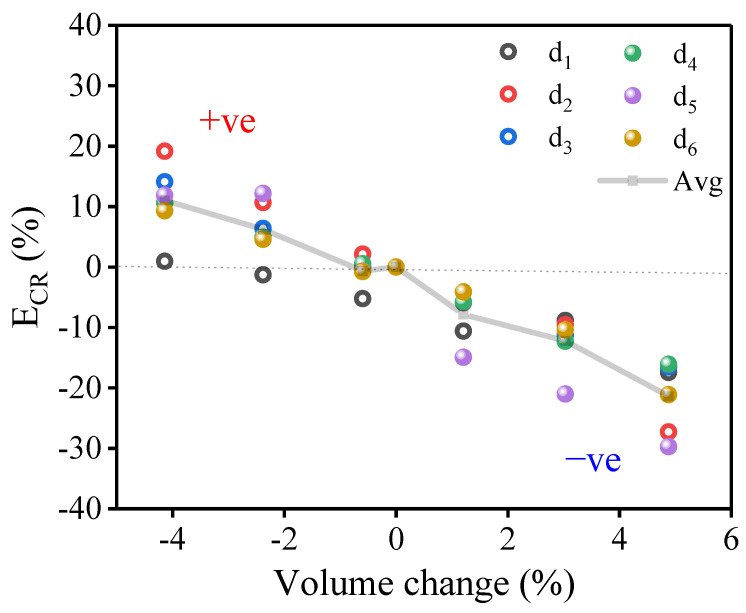
Change rate of the deformed TDE for specific directions in Ta: (d_1_) [1 1 1], (d_2_) [1 1 0], (d_3_) [1 0 0], (d_4_) [3 2 1], (d_5_) [1.12 3.8 − 0.5], (d_6_) [−0.441 −0.086 −0.894]. Each point represents the average of 20 simulations of different vibrational timings.

**Figure 7 ijms-24-03289-f007:**
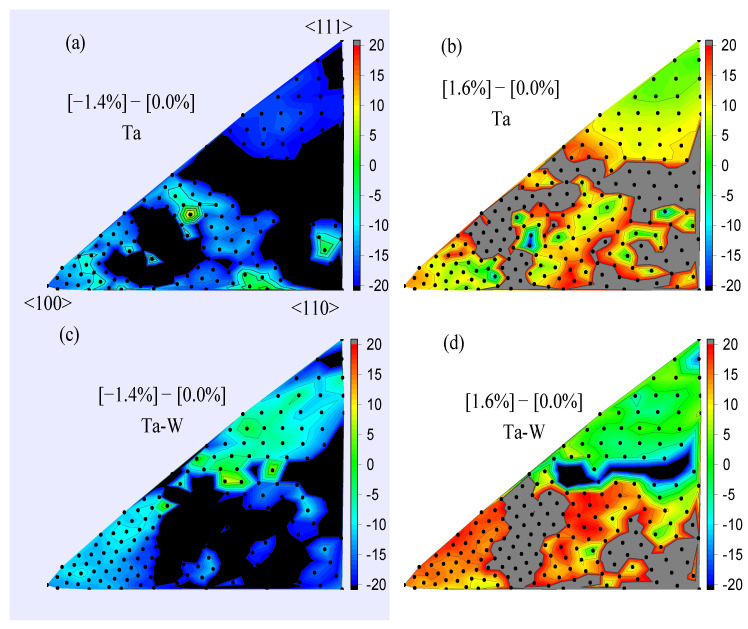
The change rate of deformed threshold displacement at 30 K. The graph shows the hydrostatic compression strain on (**a**) Ta (**c**) Ta–W and the hydrostatic tensile strain on (**b**) Ta (**d**) Ta–W.

## Data Availability

The calculations in this work were carried out using LAMMPS package and Quantum ESPRESSO as implemented in Materials Square (MatSQ). The MatSQ structure’s designing tools were used in this work. (Available at: https://www.materialssquare.com/ (accessed on 9 August 2022)).

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
