# Peer review of "Atomistic Study for the Tantalum and Tantalum–Tungsten Alloy Threshold Displacement Energy under Local Strain"

_ijms, 2023, doi:10.3390/ijms24043289_

Round 1

Reviewer 1 Report

First of all, I’m really surprised that this paper is submitted to section Molecular Biophysics (MB). The paper discusses physical and engineering aspects of radiation damage in materials which by itself has no connection to molecular biophysics whatsoever. But, of course, it’s not up to me to decide whether this submission is suitable for MB section. Apparently, the editors have decided that it is.

I think this work may potentially be suitable for publication, but in its present form it is not.

1. The paper is mainly about threshold displacement energy (TDE). However, neither its definition nor a discussion of the importance of its proper knowledge is given. All I could see in Introduction is ``To measure the radiation damage and defect production of specific materials, the threshold displacement energy (TDE) must be properly calculated” and ``The formation of a stable Frenkel pair (FP), comprising a pair of self-interstitial atoms (SIA) and vacancy, requires an energy greater than the TDE.” That’s basically all, which is unacceptable for a paper in which this is the main subject of research. Much more relevant information must be added.

2. Likewise, very little (basically, no) detail of molecular dynamics (MD) simulations is given in Methods. All I could see there is ``All MD simulations are performed using the Large-scale Atomic/Molecular Massively Parallel Simulator (LAMMPS) code [32] integrated in materials-square platform [33].’’ More adequate information must be provided: the form of interatomic potentials, the relevant parameter values, etc. Perhaps only the DFT sections provides relevant information and can remain as is.

3. The paper misses numerous important references. I will mention some of them below.

4. Now, the results themselves. I’ve always known that the values of TDE for Ta and W which are discussed in this work are about 30-35 and 50-55 eV, respectively, and I was really surprised to discover that still some researchers use values almost twice as high, and present new results which support the use of such high values. Specifically, for Ta this work suggests 60 eV, and 90 eV for W from the previous work.  Those values are about a factor of 2 higher than what I’m used to know. The authors should explain why their values are much higher than those currently adopted in the community. The reason may be unreliable interatomic potential that they used for MD, or the methodology of their determination of the values of TDE, etc. Without the clear understanding of why there is such a big difference between the accepted TDEs and the authors ones this paper cannot be published as it will cause more questions and answers.

The story about this discrepancy is well know in the community. The value of 90 eV as TDE of W was first proposed in 1981, in a conference paper based on inaccurate data derived from MD, already after the experimental data by Maury et al. were published in 1978. Based on those data, it was shown by Mason et al. (J. Phys. Cond. Mat. 26, 375701 (2014)), upon properly averaging over the entire solid angle of 4π, i.e., over all the directions of the initial velocity of the primary knock-on atom, that for W the value of TDE is 55.3 eV. Similarly, for Ta the experimental data yield the value of TDE of 22, 33±1 or 32±2 eV (Saile, Phys. Stat. Sol. A 89, K143 (1985) and references therein) and 32±3 eV (Youngblood et al., Phys. Rev. 188, 1101 (1969)). That it, for Ta, the value of TDE is ~ 30-35 eV, twice as low as that presented by the authors. More recent results on TDE for W are 48 eV from ab initio quantum MD (AIMD) and 36 eV from classical MD (CMD) (Yang and Olsson, Phys. Rev. Materials 5, 073602 (2021)), 53±8 eV from CMD (Fikar and Schäublin, J. Nucl. Mater. 386-388, 97 (2009)), and 50-55 eV from both CMD and AIMD (de Backer et al., Phys. Scripta T167, 014018 (2016)). In the last example, the values of TDE for different crystal orientations were obtained: 40/37±6 eV for <100>, 63/61±10 eV for <110>, and 44/55±10 eV for <111> from AIMD/CMD, respectively, which, upon averaging over all the directions, as in Mason et al., leads to ~ 50-55 eV as the value of TDE for W. Thus, all the recent experiments/calculations for W, including MD calculations like those discussed in the submission (but perhaps using different software and different interatomic potentials), give ~ 50-55 eV as the value of TDE for W, and the somewhat older experiments give ~ 30-35 eV as that for Ta. Both values are about twice as low as the values presented in this work. Without a clear understanding as for the nature of such a big difference the paper cannot be recommended for publication.

5. Finally, after the nature of the difference discussed above is clarified, the authors must present analytic form which describes their results on TDE for Ta-W alloy (up to 30% W, Fig. 3) and extrapolates smoothly to pure W. This form will be useful for further research on this subject.

Thus, my current recommendation is ``major revision;’’ I assume the authors could adequately address all the points of my review and come up with the revised version. Then I will be willing to review the revised version and perhaps to reconsider my opinion.

Author Response

We would like to extend our appreciation to the reviewer for their  insightful review of our manuscript. We hope that this review will contribute to the enhancement of our manuscript and provide valuable insights for future readers. Thank you for your time and effort.

Please kindly refer to the response for the pointed comments.

Kind regards

Reviewer 2 Report

By employing molecular dynamics simulations, the authors conduct a study on the threshold displacement energy of pure tantalum and tantalum-tungsten alloy under tensile and compressive strain. Different directions are also considered. The results are interesting. However, a revision needs to be done before consideration for publication in International Journal of Molecular Sciences:

1) For Ta-W alloy, a previous study (doi.org/10.1016/j.jnucmat.2021.153162) investigated the W-based W-Ta alloy. Why did the authors choose Ta-based Ta-W alloy?

2) For strain effects on the TDE, similar findings are reported in Ref. 31. What is new in this study? How the new results are different from already published ones?

3) There are some conspicuous errors in formulas, such as formulas in Line 155 and 236. The authors are suggested to check the whole manuscript carefully.

4) The settings of figures are not unified. For instance, the font and size of labels in Fig. 3 and Fig. 5 are significantly different.

5) The format of references needs to be checked such as the DOI of Refs. 3, 4, 20 and 40.

Author Response

We would like to extend our appreciation to the reviewer for their  insightful review of our manuscript. We hope that this review will contribute to the enhancement of our manuscript and provide valuable insights for future readers. Thank you for your time and effort.

Please kindly refer to the response report,

Kind regards

Round 2

Reviewer 1 Report

I'm somewhat satisfied with the authors' response of the criticism of my first review. I do think the authors have adequately addressed most of the points of the first review, yet some points didn't get proper attention, such as an analytic form that would interpolate smoothly between the TDE values for pure tungsten and pure tantalum such that to reproduce the authors' results for the Ta-W alloy is not presented in the revised version, etc. Nonetheless, I believe the paper contains enough new information to be of interest to the potential reader that warrants its acceptance and publication in this journal in its present form. So, I recommend its acceptance.